# Data reuse and the open data citation advantage

Heather A. Piwowar[1,2] and Todd J. Vision[1,2,3]

[1] National Evolutionary Synthesis Center, Durham, NC, USA
[2] Department of Biology, Duke University, Durham, NC, USA
[3] Department of Biology, University of North Carolina - Chapel Hill, Chapel Hill, NC, USA

Corresponding author
Heather A. Piwowar,
hpiwowar@gmail.com

## ABSTRACT

**Background.** Attribution to the original contributor upon reuse of published data is important both as a reward for data creators and to document the provenance of research findings. Previous studies have found that papers with publicly available datasets receive a higher number of citations than similar studies without available data. However, few previous analyses have had the statistical power to control for the many variables known to predict citation rate, which has led to uncertain estimates of the "citation benefit". Furthermore, little is known about patterns in data reuse over time and across datasets.

**Method and Results.** Here, we look at citation rates while controlling for many known citation predictors and investigate the variability of data reuse. In a multi-variate regression on 10,555 studies that created gene expression microarray data, we found that studies that made data available in a public repository received 9% (95% confidence interval: 5% to 13%) more citations than similar studies for which the data was not made available. Date of publication, journal impact factor, open access status, number of authors, first and last author publication history, corresponding author country, institution citation history, and study topic were included as covariates. The citation benefit varied with date of dataset deposition: a citation benefit was most clear for papers published in 2004 and 2005, at about 30%. Authors published most papers using their own datasets within two years of their first publication on the dataset, whereas data reuse papers published by third-party investigators continued to accumulate for at least six years. To study patterns of data reuse directly, we compiled 9,724 instances of third party data reuse via mention of GEO or ArrayExpress accession numbers in the full text of papers. The level of third-party data use was high: for 100 datasets deposited in year 0, we estimated that 40 papers in PubMed reused a dataset by year 2, 100 by year 4, and more than 150 data reuse papers had been published by year 5. Data reuse was distributed across a broad base of datasets: a very conservative estimate found that 20% of the datasets deposited between 2003 and 2007 had been reused at least once by third parties.

**Conclusion.** After accounting for other factors affecting citation rate, we find a robust citation benefit from open data, although a smaller one than previously reported. We conclude there is a direct effect of third-party data reuse that persists for years beyond the time when researchers have published most of the papers reusing their own data. Other factors that may also contribute to the citation benefit are considered. We further conclude that, at least for gene expression microarray data, a

substantial fraction of archived datasets are reused, and that the intensity of dataset reuse has been steadily increasing since 2003.

## INTRODUCTION

Sharing information facilitates science. Publicly sharing detailed research data – sample attributes, clinical factors, patient outcomes, DNA sequences, raw mRNA microarray measurements – with other researchers allows these valuable resources to contribute far beyond their original analysis. In addition to being used to confirm original results, raw data can be used to explore related or new hypotheses, particularly when combined with other publicly available data sets. Real data is indispensable when investigating and developing study methods, analysis techniques, and software implementations. The larger scientific community also benefits: sharing data encourages multiple perspectives, helps to identify errors, discourages fraud, is useful for training new researchers, and increases efficient use of funding and patient population resources by avoiding duplicate data collection.

Making research data publicly available also has challenges and costs. Some costs are borne by society: For example, data archives must be created and maintained. Many costs, however, are borne by the data-collecting investigators: Data must be documented, formatted, and uploaded. Investigators may be afraid that other researchers will find errors in their results, or "scoop" additional analyses they have planned for the future.

Personal incentives are important to balance these personal costs. Scientists report that receiving additional citations is an important motivator for publicly archiving their data (*Tenopir et al., 2011*).

There is evidence that studies that make their data available do indeed receive more citations than similar studies that do not (*Gleditsch, Metelits & Strand, 2003*; *Piwowar, Day & Fridsma, 2007*; *Ioannidis et al., 2009*; *Pienta, Alter & Lyle, 2010*; *Henneken & Accomazzi, 2011*; *Sears, 2011*; *Dorch, 2012*). These findings have been referenced by new policies that encourage and require data archiving (e.g., *Rausher et al., 2010*), demonstrating the appetite for evidence of personal benefit.

In order for journals, institutions and funders to craft good data archiving policy, it is important to have an accurate estimate of the citation differential. Estimating an accurate citation differential is made difficult by the many confounding factors that influence citation rate. In past studies, it has seldom been possible to adequately control these confounders statistically, much less experimentally. Here, we perform a large multivariate analysis of the citation differential for studies in which gene expression microarray data either was or was not made available in a public repository.

Estimating the citation differential is not enough: crafting good data archiving policy requires an understanding of its origins. How quickly do data reuse citations accrue? Do the additional citations arise due to data reuse – as we might expect – or simply from increased exposure or trust in the original study? How often do data reuse studies attribute data from more than one source?

Examining data reuse patterns on a large scale is difficult because it is difficult to automatically isolate reuse that has been attributed through a citation from citations made for other purposes. In this study we approach this issue in two ways. First, we conduct a small-scale manual review of citation contexts to understand the proportion of citations that are made in the context of data reuse. Second, we use attribution through mentions of data accession numbers, rather than citations, to explore patterns in data reuse on a much larger scale.

We seek to improve on prior work in two key ways. First, the sample size of this analysis is large – over two orders of magnitude larger than the first citation study of gene expression microarray data (*Piwowar, Day & Fridsma, 2007*), giving us the statistical power to account for a larger number of cofactors in the analyses. Thus, the resulting estimates isolate the association between data availability and citation rate with more accuracy. Second, this report goes beyond citation analysis to include analysis of data reuse attribution directly. We explore how data reuse patterns change over both the lifespan of a data repository and the lifespan of a dataset, as well as examine the distribution of reuse across datasets in a repository.

## MATERIALS AND METHODS

The primary analysis in this paper addresses the citation count of a gene expression microarray experiment relative to availability of the experiment's data, accounting for a large number of potential confounders.

### Relationship between data availability and citation

#### *Data collection*

To begin, we needed to identify a sample of studies that had generated gene expression microarray data in their experimental methods. We used a sample that had been collected previously (*Piwowar, 2011a*; *Piwowar, 2011b*); briefly, a full-text query uncovered papers that described wet-lab methods related to gene expression microarray data collection. The full-text query had been characterized as having high precision (90%, with a 95% CI of 86% to 93%) and moderate recall (56%, CI of 52% to 61%) for this task. Running the query in PubMed Central, HighWire Press, and Google Scholar identified 11603 distinct gene expression microarray papers published between 2000 and 2009.

Citation counts for 10,555 of these papers were found in Scopus and exported in November 2011. Although Scopus now has an API that would facilitate easy programmatic access to citation counts, at the time of data collection the authors were not aware of any methods for querying and exporting data other than through the Scopus website. The Scopus website limited length of a query and the number of citations that could be

exported at once. To work within these restrictions we concatenated 500 PubMed IDs at a time into 22 queries, each of the form **"PMID(1234) OR PMID(5678) OR ..."**.

The independent variable of interest was the availability of gene expression microarray data. Data availability had been previously determined for our sample articles in *Piwowar (2011a)*, so we directly reused that dataset. Datasets were considered to be publicly available if they were discoverable in either of the two most widely-used gene expression microarray repositories: NCBI's Gene Expression Omnibus (GEO), and EBI's ArrayExpress. GEO was queried for links to the PubMed identifiers in the analysis sample using **"pubmed_gds [filter]"** and ArrayExpress was queried by searching for each PubMed identifier in a downloaded copy of the ArrayExpress database. An evaluation of this method found that querying GEO and ArrayExpress with PubMed article identifiers recovered 77% of the associated publicly available datasets (*Piwowar & Chapman, 2010*).

### Primary analysis

The core of our analysis was a set of multivariate linear regressions to evaluate the association between the public availability of a study's microarray data and the number of citations received by the study.

To explore which variables to include in these regressions, we investigated correlations between the number of citations and a set of candidate variables. We also calculated correlations amongst all variables to investigate collinearity. We explored a subset of the 124 attributes from *Piwowar (2011a)* previously shown or suspected to correlate with citation rate. We selected covariates found to have a strong pairwise correlation (positive or negative) with citation rate, using Pearson correlations for numeric variables and polyserial correlations for binary and categorical variables. These covariates included: date of publication, journal that published the study, journal impact factor, journal citation half-life, number of articles published by the journal, journal open access policy, journal status as a core clinical journal by MEDLINE, number of authors of the study, country of the corresponding author, citation score of the institution of the corresponding author, publishing experience of the first and last author, and subject of the study itself (Table 1).

Publishing experience was characterized by the number of years since an author's first paper in PubMed, the number of papers the author had published, and the number of citations the author had received in PubMed Central, estimated using Authority Clusters. The topic of the study was characterized according to the article's Medical Subject Heading (MeSH) indexing terms assigned by the National Library of Medicine classifying the article as related to cancer, animals, or plants. For more information on study attributes see *Piwowar (2011a)*. Citation count was log transformed to be consistent with prior literature. Other count variables were square-root transformed. Continuous variables were represented with 3-part spines in the regression, using the rcs function in the R rms library.

The independent variable of data availability was represented as 0 or 1 in the regression, indicating whether or not associated data had been found in either of the two data repositories. Because citation counts were log transformed, the relationship of data availability to citation count was described with 95% confidence intervals after raising the regression coefficient to the power of $e$.

**Table 1 Univariate correlations between article attributes and number of citations.** Citations were log transformed and count variables were square root transformed. Pearson correlations were used for numeric variables and polyserial correlations for binary and categorical variables.

| Attribute | Variable name | Correlation |
|---|---|---|
| How many citations did the study receive? | nCitedBy.log | 1.00 |
| What was the impact factor of the journal that published the study? | journal.impact.factor.tr | 0.45 |
| How many citations had been made from PMC to the last author's previous papers? | last.author.num.prev.pmc.cites.tr | 0.30 |
| How many articles did the journal publish in 2008? | journal.num.articles.2008.tr | 0.25 |
| How many years had elapsed since the last author published his/her first paper? | last.author.year.first.pub.ago.tr | 0.24 |
| What was the mean citation score of the corresponding author's institution? | institution.mean.norm.citation.score | 0.24 |
| How many citations had been made from PMC to the first author's previous papers? | first.author.num.prev.pmc.cites.tr | 0.24 |
| How many of the journal's studies were identified as having created microarray data? | journal.microarray.creating.count.tr | 0.23 |
| How many years had elapsed since the first author published his/her first paper? | first.author.year.first.pub.ago.tr | 0.22 |
| Was the corresponding author's address in the USA? | country.usa | 0.18 |
| How many authors did the study have? | num.authors.tr | 0.17 |
| Was the study published in a journal considered a core clinical journal by MEDLINE? | pubmed.is.core.clinical.journal | 0.17 |
| How many previous papers had the last author published? | last.author.num.prev.pubs.tr | 0.15 |
| Did the study involve human subjects? | pubmed.is.humans | 0.08 |
| Was the study funded by the NIH? | pubmed.is.funded.nih | 0.07 |
| Was the study funded by an R-grant from the NIH? | has.R.funding | 0.07 |
| Did the study involve plants? | pubmed.is.plants | 0.07 |
| How many previous papers had the first author published? | first.author.num.prev.pubs.tr | 0.06 |
| Did the study involve cancer? | pubmed.is.cancer | 0.06 |
| How many cumulative years of NIH funding did the study receive? | nih.cumulative. years.tr | 0.03 |
| Was the corresponding author's address in the UK? | country.uk | 0.03 |
| How many NIH grants did the study receive? | num.grants.via.nih.tr | 0.02 |
| What was the sum of the annual grants received from the NIH? | nih.sum.avg.dollars.tr | 0.01 |
| Did the study involve bacteria? | pubmed.is.bacteria | 0.01 |
| Was an associated dataset found in GEO or ArrayExpress? | dataset.in.geo.or.ae | 0.01 |
| How many of the last author's previous papers were identified as creating gene expression microarray data? | last.author.num.prev.microarray.creations.tr | 0.01 |
| Did the study use cultured cells? | pubmed.is.cultured.cells | −0.01 |
| How many of the first author's previous papers were identified as creating gene expression microarray data? | first.author.num.prev.microarray.creations.tr | −0.01 |
| Was this study listed as one that had reused data from GEO? | pubmed.is.geo.reuse | −0.01 |
| Was the corresponding author's institution a government institution? | institution.is.govnt | −0.01 |
| Was the corresponding author's address in Australia? | country.australia | −0.02 |
| Did the study receive interamural NIH funding? | pubmed.is.funded.nih.intramural | −0.03 |
| Was the corresponding author's address in Canada? | country.canada | −0.05 |
| What is the rank of the corresponding author's institution? | institution.rank | −0.06 |
| Was the last author female? | last.author.female | −0.07 |
| Was the first author female? | first.author.female | −0.08 |
| Was the corresponding author's address in Japan? | country.japan | −0.10 |
| Did the study involve animals? | pubmed.is.animals | −0.11 |
| Was the corresponding author's address in China? | country.china | −0.19 |
| Was the corresponding author's address in Korea? | country.korea | −0.26 |
| Was the journal that published the study considered an open access journal? | pubmed.is.open.access | −0.30 |
| What year was the study published? | pubmed.year.published | −0.58 |
| What date was the study published? | pubmed.date.in.pubmed | −0.59 |

### Comparison to 2007 study

We ran two modified analyses to attempt to reproduce the findings of *Piwowar, Day & Fridsma (2007)* using the larger dataset of the current study. First, we used a subset of studies with roughly the same inclusion criteria as the earlier paper – studies on cancer in humans, published prior to 2003 – and the same regression coefficients: publication date, impact factor, and whether the corresponding author's address is in the USA. We followed that with a second regression that included several additional important covariates: number of authors and number of previous citations by the last author.

### Stratification by year

Because publication date is a strong correlate with both citation rate and data availability, we performed a separate analysis stratifying the sample by publication year, in addition to including publication date as a covariate. Fewer covariates could be included in these yearly regressions because they included fewer datapoints than the full regression. The yearly regressions included date of publication, journal that published the study, journal impact factor, journal's open access policy, number of authors of the study, citation score of the institution of the corresponding author, previous number of PubMed Central citations received by the first and last author, whether the topic was cancer, and whether the study used animals.

### Manual review of citation context

We manually reviewed the context of citations to data collection papers to estimate how many citations to data collection papers were made in the context of data reuse. We (Jonathan Carlson, in acknowledgements) reviewed 138 citations chosen randomly from the set of all citations to 100 data collection papers. Specifically, we randomly selected 100 datasets deposited in GEO in 2005. For each dataset, we located the data collection article within Thomson Reuter's Web of Science based on its title and authors, and exported the list of all articles that cited this data collection article. From this, we selected random citations stratified by the total number of times the data collection article had been cited. By manual review of the relevant full-text of each paper, we determined whether the data from the associated dataset had been reused within the study.

Web of Science was used to identify citations for this step rather than the Scopus citation database used in previous steps. This is because extracting citations in this step did not require the (at the time) Scopus-only feature of searching by PubMed ID and the investigators had access to the Web of Science through an institutional subscription.

## Data reuse patterns from accession number attribution

A second, independent dataset was collected to correlate with reuse attributions made through mentions of accession numbers rather than formal citations.

### Data collection

Datasets are sometimes attributed directly through mention of the dataset identifier (or accession number) in the full-text, in which case the reuse may not contribute to the citation count of the original paper. To capture these instances of reuse, we collected a

separate dataset to study reuse patterns based on direct data attribution. We used the NCBI eUtils library and custom Python code to obtain a list of all datasets deposited into the Gene Expression Omnibus data repository, then searched PubMed Central for each of these dataset identifiers (using queries of the form "'GSEnnnn' OR 'GSE nnnn'"). For each hit we recorded the PubMed Central ID of the paper that mentioned the accession number, the year of paper publication, and the author surnames. We also recorded the dataset accession number, the year of dataset publication, and the investigator names associated with the dataset record.

### Statistical analysis

To focus on data reuse by third party investigators (rather than authors attributing datasets they had collected themselves), we excluded papers with author surnames matching those of authors who deposited the original dataset, as in *Piwowar, Carlson & Vision (2011a)*. PubMed Central contains only a subset of papers recorded in PubMed. As described in *Piwowar, Carlson & Vision (2011a)*, to extrapolate from the number of data reuses in PubMed Central to all possible data reuses in PubMed, we divided the yearly number of hits by the ratio of papers in PMC to papers in PubMed for this domain (domain was measured as the number of articles indexed with the MeSH term "gene expression profiling").

We retained papers published between 2001 and 2010 as reuse candidates. We excluded 2011 because it had a dramatically lower proportion of papers in PubMed Central at the time of our data collection: the NIH requirement to deposit a paper into PMC permits a 12 month embargo.

To understand our findings on a per-dataset basis, we stratified reuse estimates by year of dataset submission and normalized our reuse findings by the number of datasets deposited that year.

## Data and script availability

Statistical analyses were last run on Wednesday, April 3, 2013 with R version 2.15.1 (2012-06-22). Packages used included reshape2 (*Wickham, 2007*), plyr (*Wickham, 2011*), rms (*Harrell, 2012*), polycor (*Fox, 2010*), ascii (*Hajage, 2011*), ggplot2 (*Wickham, 2009*), gplots (*Warnes et al., 2012*), knitr (*Xie, 2012*), and knitcitations (*Boettiger, 2013*). *P*-values were two-tailed.

Raw data and statistical scripts are available in the Dryad data repository at http://doi.org/10.5061/dryad.781pv. Data collection scripts are on GitHub at https://github.com/hpiwowar/georeuse and https://github.com/hpiwowar/pypub.

The Markdown version of this manuscript with interleaved statistical scripts (*Xie, 2012*) is on GitHub https://github.com/hpiwowar/citation11k. Publication references are available in a publicly-available Mendeley group to facilitate exploration at http://www.mendeley.com/groups/2223913/11k-citation/papers/.

**Table 2** Proportion of sample published in most common journals.

| | | |
|---|---|---|
| 1 | Cancer Res | 0.04 |
| 2 | Proc Natl Acad Sci USA | 0.04 |
| 3 | J Biol Chem | 0.04 |
| 4 | BMC Genomics | 0.03 |
| 5 | Physiol Genomics | 0.03 |
| 6 | PLoS One | 0.02 |
| 7 | J Bacteriol | 0.02 |
| 8 | J Immunol | 0.02 |
| 9 | Blood | 0.02 |
| 10 | Clin Cancer Res | 0.02 |
| 11 | Plant Physiol | 0.02 |
| 12 | Mol Cell Biol | 0.01 |

**Table 3** Proportion of sample published each year.

| 2001 | 2002 | 2003 | 2004 | 2005 | 2006 | 2007 | 2008 | 2009 |
|---|---|---|---|---|---|---|---|---|
| 0.02 | 0.05 | 0.08 | 0.11 | 0.13 | 0.12 | 0.17 | 0.18 | 0.15 |

## RESULTS

### Description of cohort

We identified 10,557 articles published between 2001 and 2009 as collecting gene expression microarray data. Publicly available datasets in GEO or ArrayExpress had been found for 2,617 of these articles (25%, 95% confidence interval 24% to 26%). The papers were published in 667 journals, with the top 12 journals accounting for 30% of the papers (Table 2). Microarray papers were published more frequently in later years: 2% of articles in our sample were published in 2001, compared to 15% in 2009 (Table 3). The papers were cited between 0 and 2,643 times, with an average of 32 citations per paper and a median of 16 citations.

### Data availability is associated with citation benefit

Without accounting for any confounding factors, the distribution of citations was similar for papers with and without archived data. That said, we hasten to mention several strong confounding factors. For example, the number of citations a paper has received is strongly correlated to the date it was published: older papers have had more time to accumulate citations. Furthermore, the probability of data archiving is also correlated with the age of an article – more recent articles are more likely to archive data (*Piwowar, 2011a*). Accounting for publication date, the distribution of citations for papers with available data is right-shifted relative to the distribution for those without, as seen in Fig. 1.

Other variables have been shown to correlate with citation rate (*Fu & Aliferis, 2008*). Because single-variable correlations can be misleading, we performed multivariate

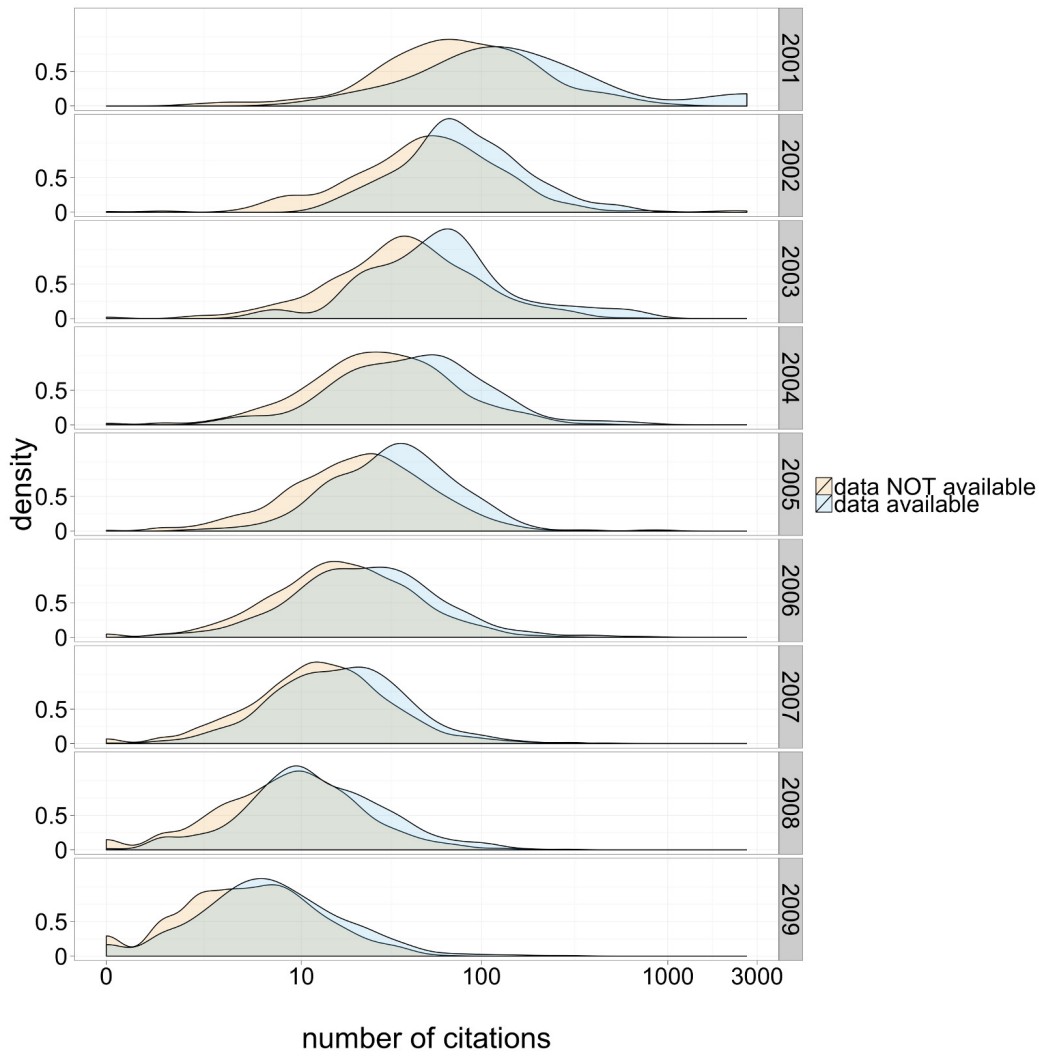

**Figure 1 Citation density for papers with and without publicly available microarray data, by year of study publication.**

regression to isolate the relationship between data availability and citation rate from confounders.

The multivariate regression included attributes representing an article's journal, journal impact factor, date of publication, number of authors, number of previous citations of the first and last author, number of previous publications of the last author, whether the paper was about animals or plants, and whether the data was made publicly available. Citations were 9% higher for papers with available data, independent of other variables ($p < 0.01$, 95% confidence intervals [5% to 13%]).

We also analyzed a subset of manually curated articles. The findings were similar to those of the whole sample, supporting our assumption that errors in automated inclusion criteria determination did substantially influence the estimate (see Article S1).

### More covariates led to a more conservative estimate

Our estimate of citation benefit, 9% as per the multivariate regression, is notably smaller than the 69% (95% confidence intervals of 18% to 143%) citation advantage found by *Piwowar, Day & Fridsma (2007)*, even though both studies examined publicly available gene expression microarray data. There are several possible reasons for this difference.

First, *Piwowar, Day & Fridsma (2007)* concentrated on datasets from high-impact studies: human cancer microarray trials published in the early years of microarray analysis (between 1999 and 2003). By contrast, the current study included gene expression microarray data studies on any subject published between 2001 and 2009. Second, because the *Piwowar, Day & Fridsma (2007)* sample was small (85 papers), the previous analysis included only a few covariates: publication date, journal impact factor, and country of the corresponding author.

We attempted to reproduce the *Piwowar, Day & Fridsma (2007)* methods with the current sample. Limiting the inclusion criteria to studies with MeSH terms "human" and "cancer", and to papers published between 2001 and 2003, reduced the cohort to 308 papers. Running this subsample with the covariates used in the *Piwowar, Day & Fridsma (2007)* paper resulted in a comparable estimate to the 2007 paper: a citation increase of 47% (95% confidence intervals of 6% to 103%).

The subsample of 308 papers was large enough to include a few additional covariates: number of authors and citation history of the last author. Including these important covariates decreased the estimated effect to 18% with a confidence interval that spanned a loss of 17% citations to a benefit of 66%.

### Citation benefit over time

After completing our comparison to prior results, we returned to the whole sample. Because publication date is such a strong correlate with both citation rate and data availability, we ran regressions for each publication year individually. The estimate of citation benefit varied by year of publication. The citation benefit was greatest for data published in 2004 and 2005, at about 30%. Earlier years showed citation benefits with wider confidence intervals due to relatively small sample sizes, while more recently published data showed a less pronounced citation benefit (Fig. 2).

### Data reuse is a demonstrable component of citation benefit

To estimate the proportion of the citation benefit directly attributable to data reuse, we randomly selected and manually reviewed 138 citations. We classified eight (6%) of the citations as attributions for data reuse (95% CI: 3% to 11%).

### Evidence of reuse from mention of dataset identifiers in full text

A complementary dataset was collected and analyzed to characterize data reuse: direct mention of dataset accession numbers in the full text of papers. In total there were 9274 mentions of GEO datasets in papers published between 2000 and 2010 within PubMed Central across 4543 papers written by author teams whose last names did not match the names of those who deposited the data. Extrapolating this to all of PubMed, we estimated

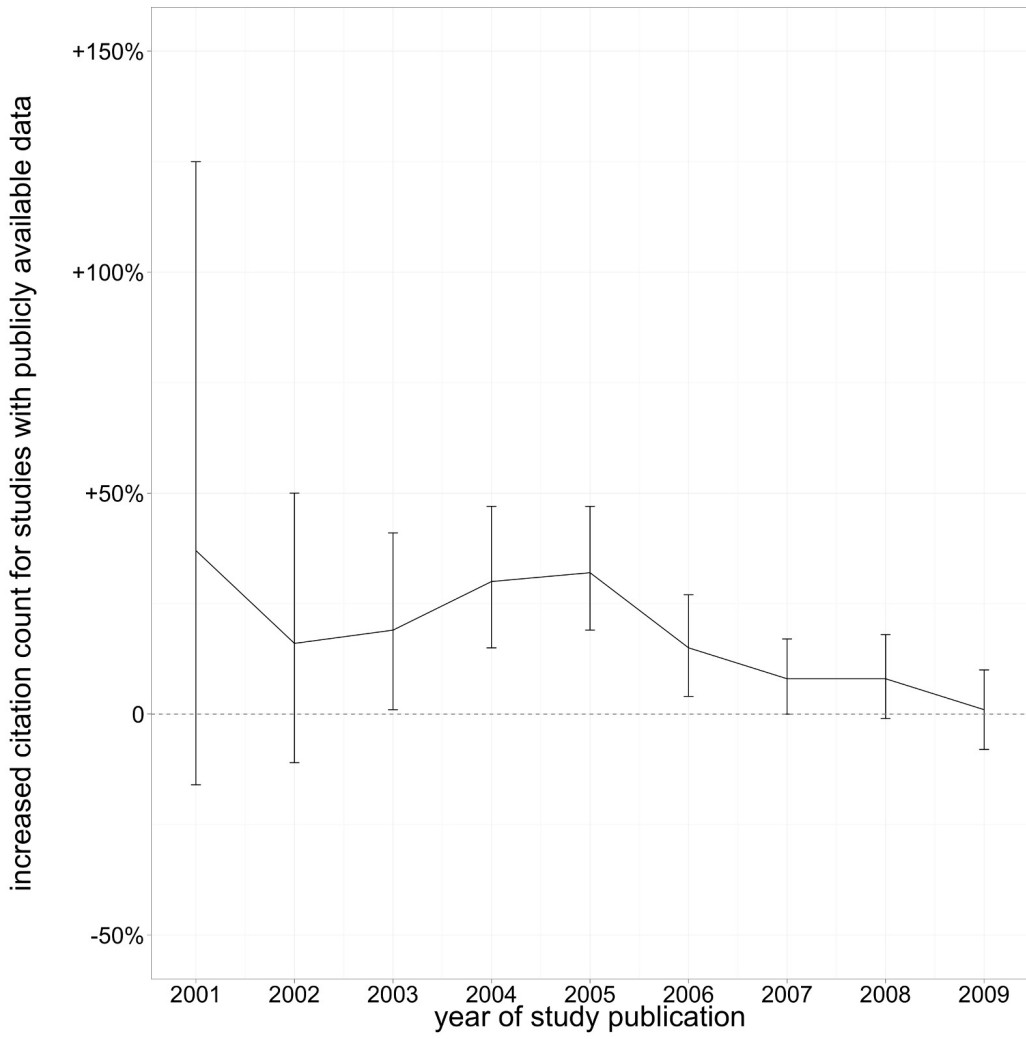

**Figure 2 Increased citation count for studies with publicly available data, by year of publication.** Estimates from multivariate analysis, lines indicate 95% confidence intervals.

there may be about $1.4081 \times 10^4$ third-party reuses of GEO data attributed through accession numbers in all of PubMed for papers published between 2000 and 2010.

The number of reuse papers started to grow rapidly several years after data archiving rate started to grow. In recent years both the number of datasets and the number of reuse papers have been growing rapidly, at about the same rate, as shown in Fig. 3. The level of third-party data use was high: for 100 datasets deposited in year 0, we estimate that 40 papers in PubMed reused a dataset by year 2, 100 by year 4, and more than 150 by year 5. This data reuse curve is remarkably constant for data deposited between 2004 and 2009. The reuse growth trend for data deposited in 2003 has been slower, perhaps because 2003 data is not as ground-breaking as earlier data, and probably not as standards-compliant and technically relevant as later data.

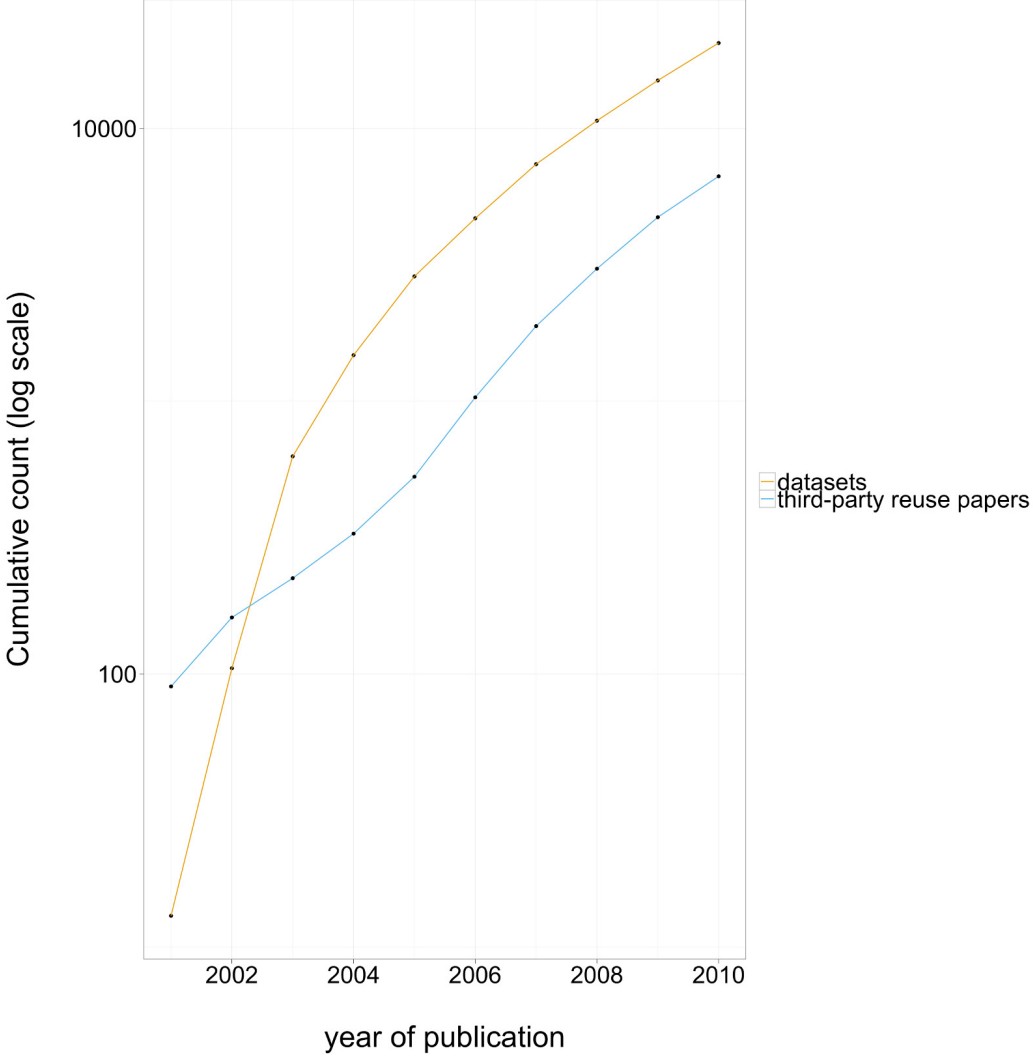

**Figure 3 Cumulative number of datasets deposited in GEO each year, and cumulative number of third-party reuse papers published that directly attribute GEO data published each year, log scale.**

We found that most instances of self-reuse (identified by surname overlap with data submission record) were published within two years of dataset publication. This pattern contrasts sharply with third party data reuse, as shown in Fig. 4. The cumulative number of third-party reuse papers is illustrated in Fig. 5; separate lines are displayed for different dataset publication years.

Because the number of datasets published has grown dramatically with time, it is instructive to consider the cumulative number of third-party reuses normalized by the number of datasets deposited each year (Fig. S1). In the earliest years for which data is available, 2001–2002, there were relatively few data deposits, but these datasets have been disproportionately reused. We excluded the early years from the plot to examine the

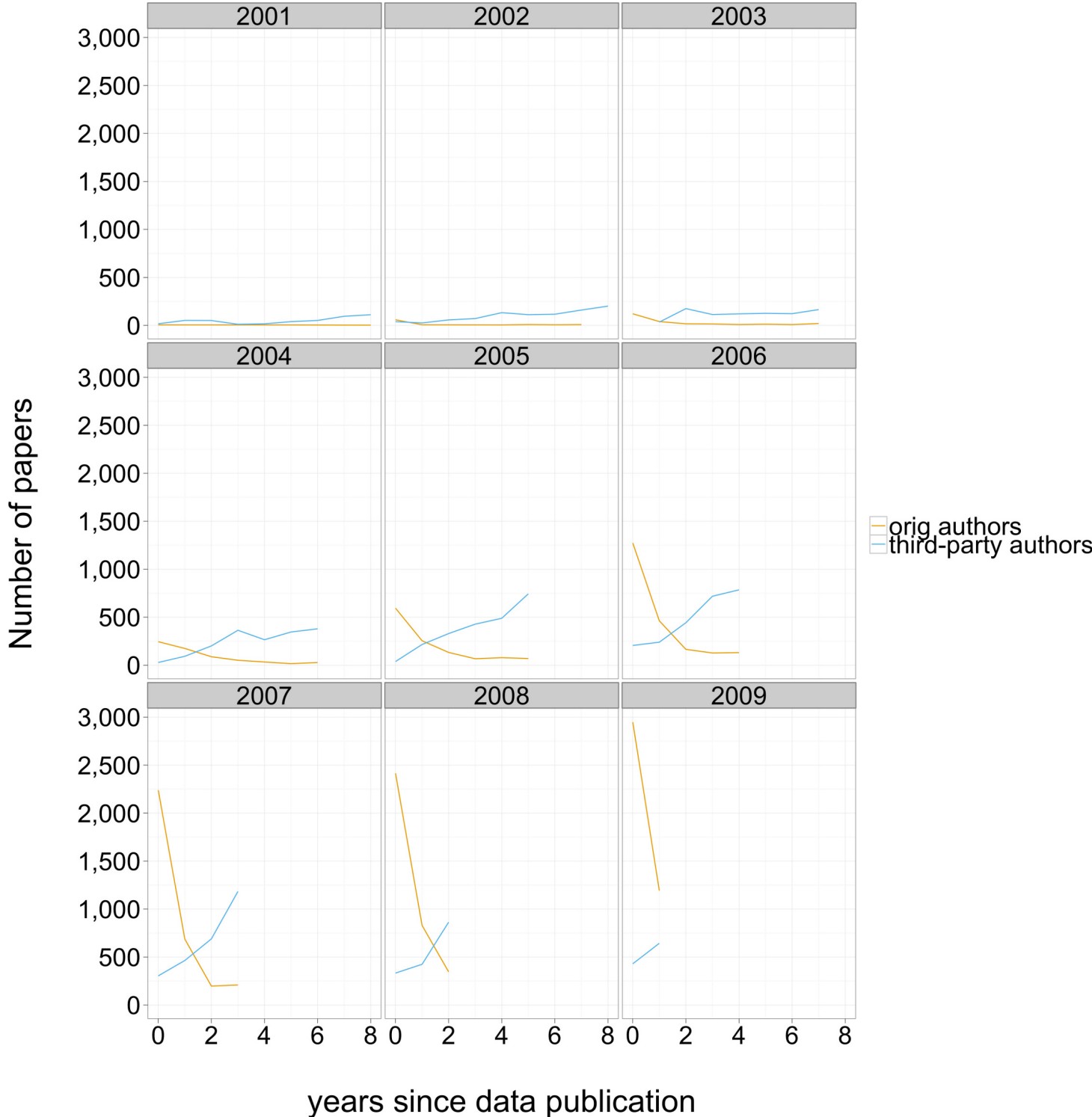

**Figure 4 Number of papers mentioning GEO accession numbers.** Each panel represents reuse of a particular year of dataset submissions, with number of mentions on the y axis, years since the initial publication on the x axis, and a line for reuses by the data collection team and a line for third-party investigators.

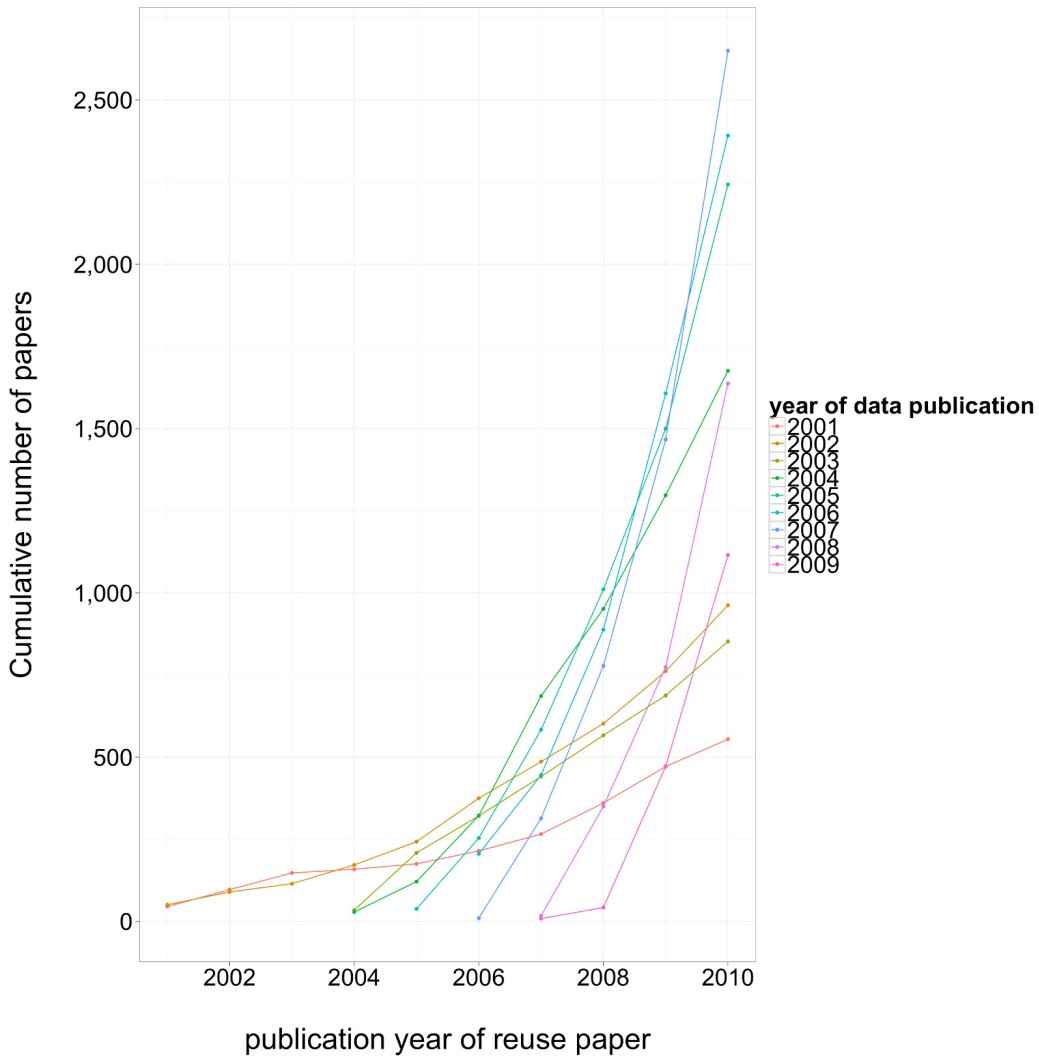

**Figure 5 Cumulative number of third-party reuse papers, by date of reuse paper publication.** Separate lines are displayed for different dataset submission years.

pattern of data reuse once gene expression datasets became more common. Since 2003, the rate at which individual datasets were reused increased with each year of data publication.

## Growth in the number of datasets in each reuse paper over time

The number of distinct datasets used in a reuse paper was found to increase over time (Fig. 6). From 2002 to 2004 almost all reuse papers only used one or two datasets. By 2010, 25% of reuse papers used 3 or more datasets.

## Distribution of reuse across datasets

It is useful to know the distribution of reuse amongst datasets. Because our methods only detect reuse by papers in PubMed Central (a small proportion of the biomedical literature) and only when the accession number is given in the full text, our estimates of

**Peer**J

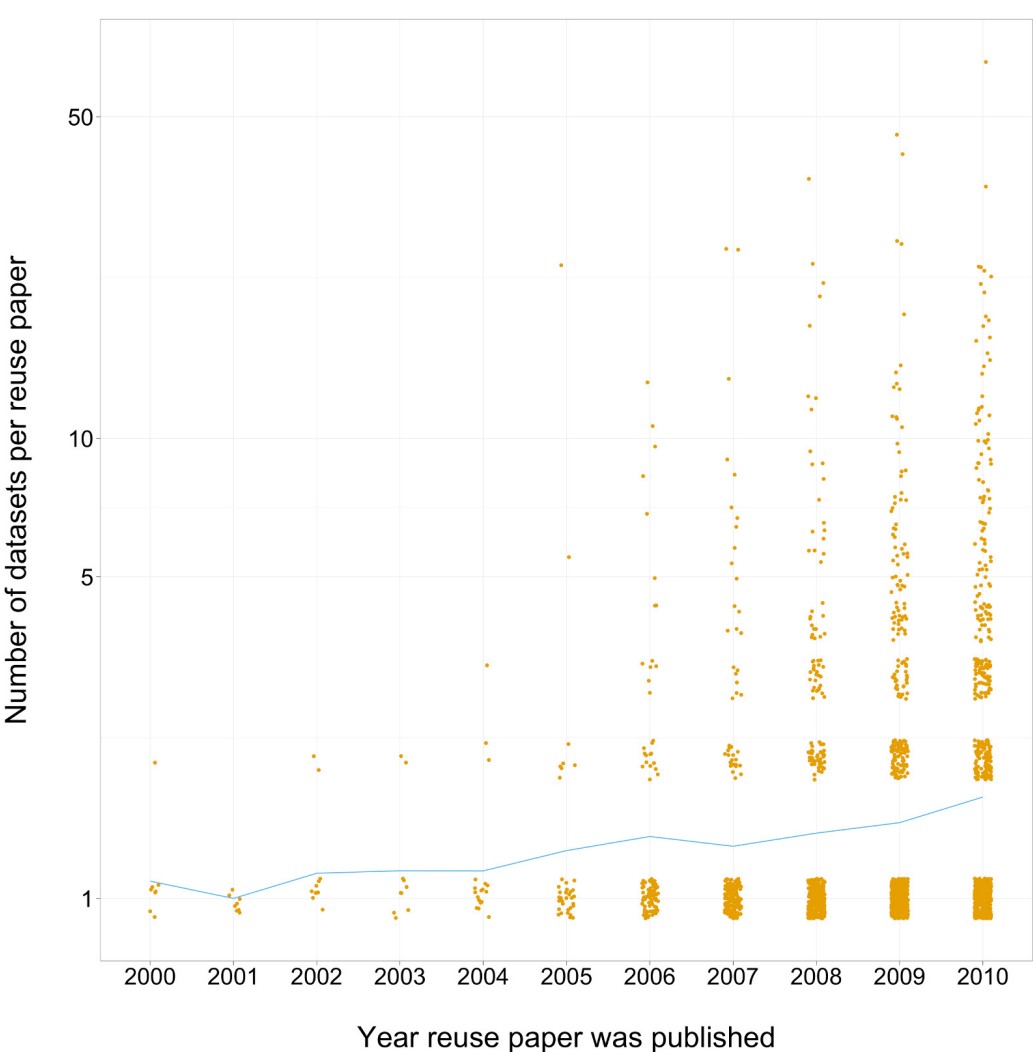

**Figure 6** Scatterplot of year of publication of third-party reuse paper (with jitter) vs number of GEO datasets mentioned in the paper (log scale). The line connects the mean number of datasets attributed in reuse papers vs publication year.

reuse are extremely conservative. Despite this, we found that reuse was not limited to only a few papers (Fig. 7). Nearly all datasets published in 2001 were reused at least once. The proportion of reused datasets declined in subsequent years, with a plateau of about 20% for data deposited between 2003 and 2007. The actual rate of reuse across all methods of attribution, and extrapolated to all of PubMed, is probably much higher.

## Distribution of the age of reused data

We found the authors of third-party data reuse papers were most likely to use data that was 3–6 years old by the time their paper was published, normalized for how many datasets were deposited each year (Fig. 8). For example, in aggregate, microarray reuse papers from 2005 mentioned the accession numbers of more than 5% of all datasets that had been submitted two years earlier, in 2003. Reuse papers from 2008 mentioned about 7% of the

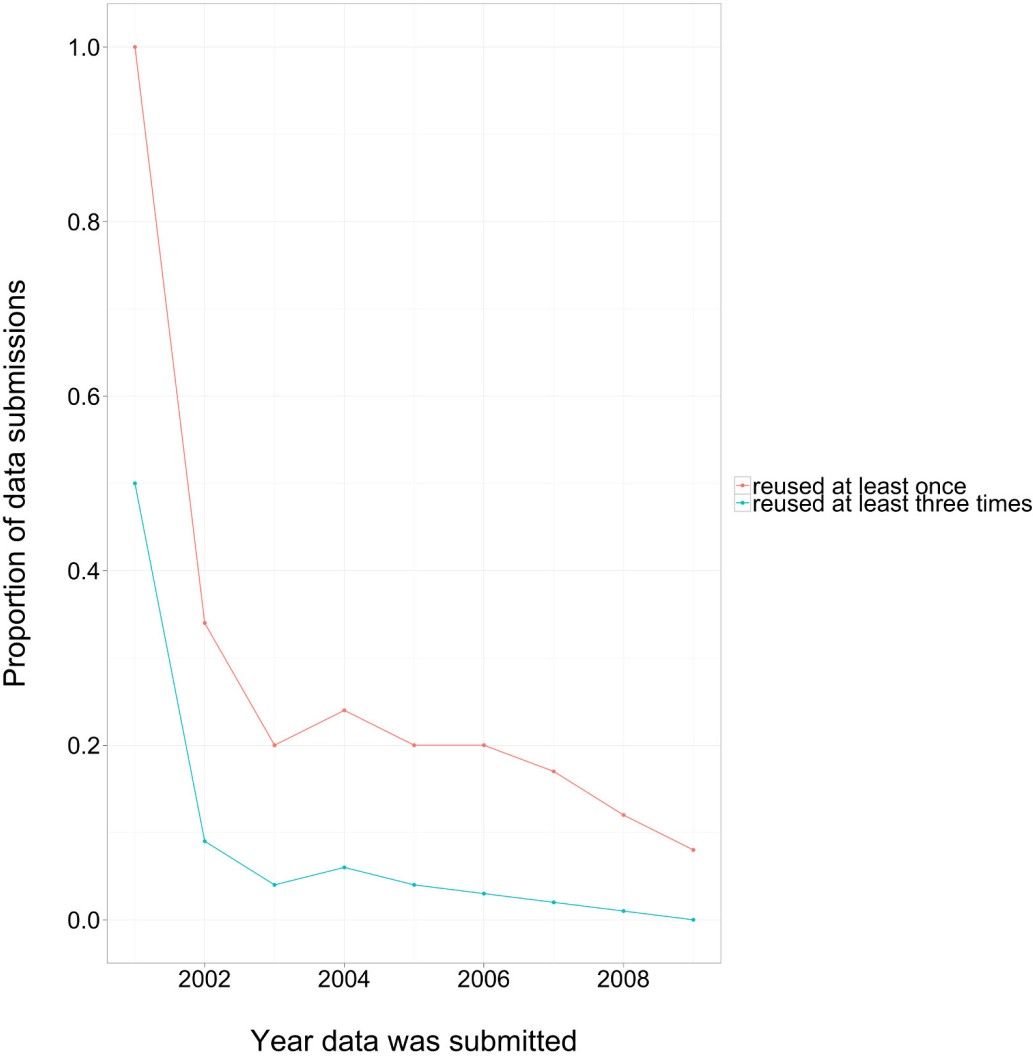

**Figure 7** **Proportion of data reused by third-party papers vs year of data submission.** These estimates are a lower bound: they only considered reuse by papers in PubMed Central, and only when reuse was attributed through direct mention of a GEO accession number.

datasets submitted two years earlier (in 2006), more than 10% of the datasets submitted 3 and 4 years previously (2005 and 2004), and about 7% of the datasets submitted 5 years earlier, in 2003.

## DISCUSSION

### The open data citation benefit

One of the primary findings of this analysis is that papers with publicly available microarray data received more citations than similar papers that did not make their data available, even after controlling for many variables known to influence citation rate. We found the open data citation benefit for this sample to be 9% overall (95% confidence

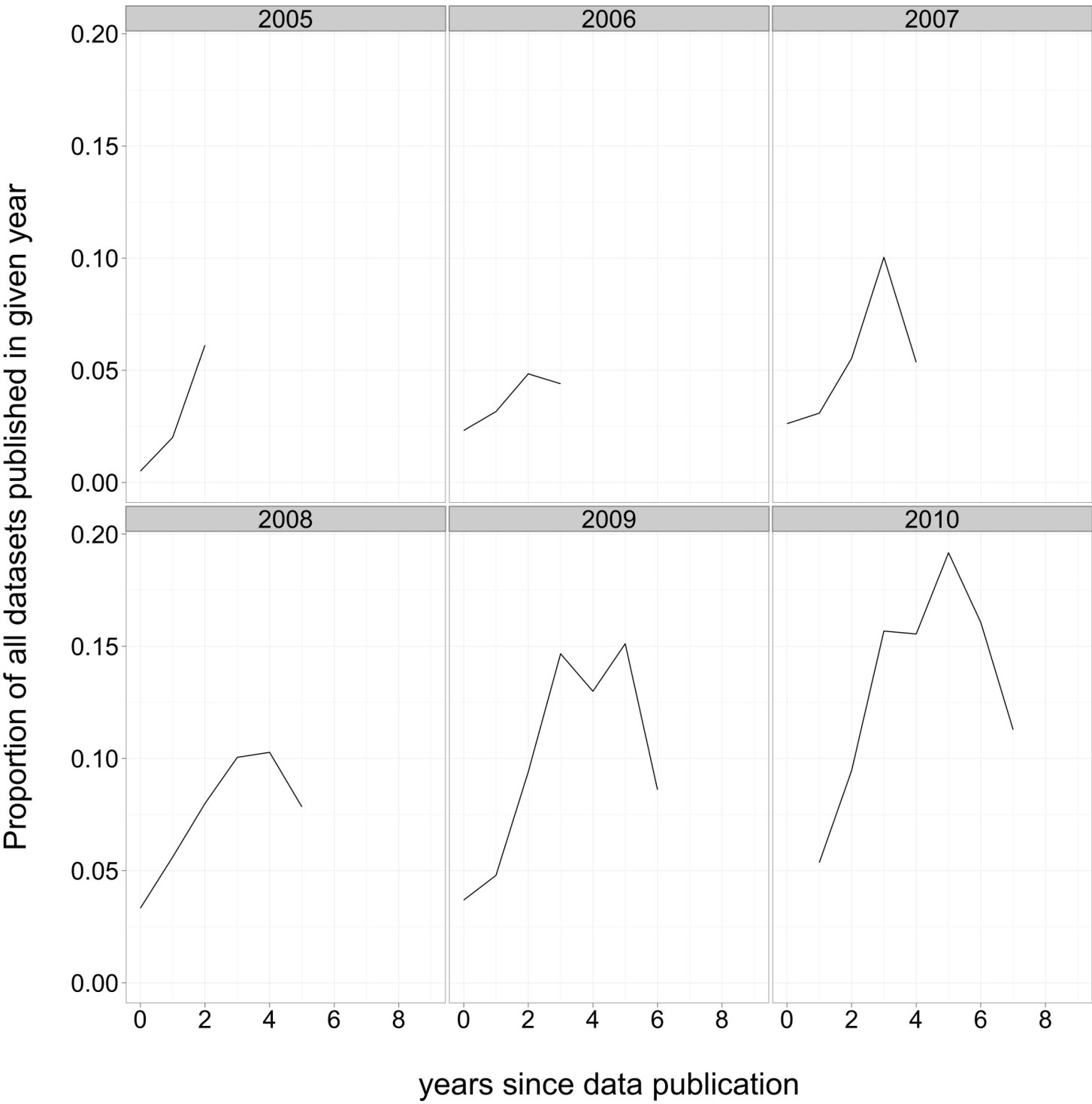

**Figure 8** **Proportion of data submissions that contributed to data reuse papers, by year of reuse paper publication and dataset submission.** Each panel includes a cohort of data reuse papers published in a given year. The lines indicate the proportion of datasets that were mentioned, in aggregate, by the data reuse papers, by the year of dataset publication. The proportion is relative to the total number of datasets submitted in a given year.

interval: 5% to 13%), but the benefit depended heavily on the year the dataset was made available. Datasets deposited very recently have so far received no (or few) additional citations, while those deposited in 2004–2005 showed a clear benefit of about 30% (confidence intervals 15% to 48%). Older datasets also appeared to receive a citation benefit, but the estimate is less precise because relatively little microarray data was collected or archived in the early 2000s.

The citation benefit reported here is smaller than that reported in the previous study by *Piwowar, Day & Fridsma (2007)*, which estimated a citation benefit of 69% for human cancer gene expression microarray studies published before 2003 (95% confidence intervals of 18% to 143%). Our attempt to replicate the *Piwowar, Day & Fridsma (2007)* study here suggests that aspects of both the data and analysis can help to explain the quantitatively different results. It appears that clinically relevant datasets released early in the history of microarray analysis had a particularly strong impact. Importantly, however, the new analysis also suggested that the previous estimate was confounded by significant citation correlates, including the total number of authors and the citation history of the last author. This finding reinforces the importance of accounting for covariates through multivariate analysis and the need for large samples to support full analysis: the 69% estimate is probably too high, even for its high-impact sample. Nonetheless, a 10%–30% is citation benefit may still be an effective motivator for data deposit, given that prestigious journals have been known to advertise their impact factors to three decimal places (*Smith, 2006*).

A paper with open data may be cited for reasons other than data reuse, and open data may be reused without citation of the original paper. Ideally, we would like to separate these two phenomena (data reuse and paper citation) and measure how often the latter is driven by the former. In our manual analysis of 138 citations to papers with open data, we observed that 6% (95% CI: 3% to 11%) of citations were in the context of data reuse. Although this methodology and the sample size do not allow us to estimate with any precision the proportion of the citation benefit attributable to data reuse, the result is consistent with data reuse being a major contributor.

Another important result of the citation analysis is that the number of papers based on self data reuse declined steeply after two years, while data reuse papers by third-party authors continued to accumulate even after six years. This finding suggests that although researchers may have some incentive for protecting their own exclusive use of data close to the time of the initial publication, the equation changes dramatically after a short period. This finding provides some evidence to guide policy decisions regarding the length of data embargoes allowed by journal archiving policies such as the Joint Data Archiving Policy described by *Rausher et al. (2010)*.

While we cannot generalize from these detailed patterns of data reuse and citation to other datatypes or domains, the cumulative citation benefit seems to be quantitatively similar in a number of different fields (*Gleditsch, Metelits & Strand, 2003*; *Piwowar, Day & Fridsma, 2007*; *Ioannidis et al., 2009*; *Pienta, Alter & Lyle, 2010*; *Henneken & Accomazzi, 2011*; *Sears, 2011*; *Dorch, 2012*).

## Challenges collecting citation data

This study required obtaining citation counts for thousands of articles using PubMed IDs. This process was not supported at the time of data collection using either Thomson Reuter's Web of Science or Google Scholar. Although this type of query was supported by Elsevier's Scopus database, we lacked institutional access to Scopus, individual subscriptions were not available, and attempts to request access through Scopus staff were unsuccessful. One of us (HP) attempted to use the British Library's walk-in access of Scopus while visiting the UK. Unfortunately, the British Library's policies did not permit any method of electronic input of the PubMed identifier list (the list is 10,000 elements long). HP eventually obtained official access to Scopus through a Research Worker agreement with Canada's National Research Library (NRC-CISTI), after being fingerprinted to obtain a police clearance certificate because she had recently lived in the United States.

Our understanding of research practice suffers because access to tools and data is so difficult.

## Patterns of data reuse

To better understand patterns of data reuse, a larger sample of reuse instances was needed than could easily be assembled through manual classification of citation context. To that end, we used a complementary source of information about reuse of the same datasets: direct mention of GEO or ArrayExpress accession numbers within the body of a full-text research article. The large number of instances of reuse identified this way allowed us to ask questions about the distribution of reuse over time and across datasets. The results indicate that dataset reuse has been increasing over time (excluding the initial years of GEO and ArrayExpress when few datasets were deposited and reuse appears to have been atypically broad). Recent reuse analyses include more datasets, on average, than older reuse studies. Also, the fact that reuse was greatest for datasets published between three and six years previously suggests that the lower citation benefit we observed for recent papers is due, at least in part, to a relatively short follow-up time.

Extrapolating to all of PubMed, we estimate the number of reuse papers published per year is on the same order of magnitude – and likely greater – than the number of datasets made available. This data reuse curve is remarkably constant for data deposited between 2004 and 2009. This finding reinforces the conclusions of an earlier analysis: even modest data reuse can provide an impressive return on investment for science funders (*Piwowar, Vision & Whitlock, 2011b*).

Finally, we observed a moderate proportion of datasets being reused by third parties (more than 20% of the datasets deposited between 2003 and 2007). It is important to recognize that this is likely a gross underestimate. It includes only those instances of reuse that can be recognized through the mention of accession number in PubMed Central. No attempt has been made to extrapolate these distribution statistics to all of PubMed, nor to identify additional attributions through paper citations or mentions of the archive name

alone. Further, many important instances of data reuse leave no trace in the published literature, such as those in education and training.

### Reasons for the data citation benefit

While we cannot exclude that the open data citation benefit is driven entirely by third-party data reuse, there may be other factors contributing to the effect either directly or indirectly. The literature that has considered the possibility of an Open Access citation benefit (e.g., *Craig et al., 2007*) indicates a number of other factors that may also be relevant to open data. Building upon this work, we suggest several possible sources for an "Open Data citation benefit":

1. *Data Reuse*. Papers with available datasets can be used in ways that papers without data cannot, and may receive additional citations as a result.
2. *Credibility Signalling*. The credibility of research findings may be higher for research papers with available data. Such papers may be preferentially chosen as background citations or the foundation of additional research.
3. *Increased Visibility*. Third party researchers may be more likely to encounter a paper with available data, either by a direct link from the data or indirectly through cross-promotion. For example, links from a data repository to a paper may increase the search ranking of the research paper.
4. *Early View*. When data is made available before a paper is published, some citations may accrue earlier than they would otherwise because of accelerated awareness of the methods, findings, and so on.
5. *Selection Bias*. Authors may be more likely to publish data for papers they judge to be their best quality work, because they are particularly proud or confident of the results (*Wicherts, Bakker & Molenaar, 2011*).

Importantly, almost all of these mechanisms are aligned with more efficient and effective scientific progress: increased data use, facilitated credibility determination, earlier access, improved discoverability, and a focus on best work through data availability are good for both investigators and the science community as a whole. Working through the one area where incentives between scientific good and author incentives conflict, – finding weaknesses or faults in published research – may require mandates. Or, instead, the research community may eventually come to associate withheld data with poor quality research, as it does today for findings that are not disclosed in a peer-reviewed paper (*Ware, 2008*).

The citation benefit observed in the current study is consistent with data reuse found in this study and the small-scale annotation reported in *Rung & Brazma (2013)*. Nonetheless, it is possible some of the other sources suggested above may have contributed citations for the studies with available data. Further work will be needed to understand the relative contributions from each source. For example, in-depth analyses of all publications from a set of data-collecting authors could support measurement of selection bias. Observing search behavior of researchers, and the returned search hit results, could characterize increased

visibility because of data availability. Hypothetical examples could be provided to authors to determine whether they would be systematically more likely to cite a paper with available data in situations in which they are considering the credibility of research findings.

## Future work

Future work could improve on these results by considering and integrating all methods of data use attribution. This holistic effort would include identifying citations to the paper that describes the data collection, mentions of the dataset identifier itself – whether in full text, the references section, or supplementary information – citations to the dataset as a first-class research object, and even mentions of the data collection investigators in acknowledgement sections. The citations and mentions would need classification based on context to ensure they are in the context of data reuse.

The obstacles encountered in obtaining the citation data needed for this study, as described earlier in the Discussion, demonstrate that improvements in tools and practice are needed to make impact tracking easier and more accurate, for day-to-day analyses as well as studies for evidence-based policy. Such research is hamstrung without programmatic access to the full-text of the research literature and to the citation databases that underpin impact assessment. The lack of conventions and tool support for data attribution (*Mooney & Newton, 2012*) is also a significant obstacle, undoubtedly leading to undercounting in the present study. There is much room for improvement, and we are hopeful about recent steps toward data citation standards taken by initiatives such as DataCite.

Data from current and future studies could begin to be used to estimate the impact of policy decisions. For example, do embargo periods decrease the level of data reuse? Do restrictive or poorly articulated licensing terms decrease data reuse? Which types of data reuse are facilitated by robust data standards and which types are unaffected?

Qualitative assessment of data reuse is an essential complement to large-scale quantitative analyses. Repeating and extending previous studies will help develop an understanding of the potential of data reuse, areas of progress, and remaining challenges (e.g., *Zimmerman, 2003*; *Wan & Pavlidis, 2007*; *Wynholds et al., 2012*; *Rolland & Lee, 2013*). Usage statistics from primary data repositories and value-added repositories are also useful sources of insight into reuse patterns (*Rung & Brazma, 2013*).

Citations are blind to many important types of data reuse. The impact of data on practitioners, educators, data journalists, and industry researchers is not captured by attribution patterns in the scientific literature. Altmetrics indicators reveal discussions in social social media, syllabi, patents, and theses: analyzing such indicators for datasets would provide valuable evidence of reuse beyond the scientific literature. As evaluators move away from assessing research based on journal impact factor and toward article-level metrics, post-publication metrics rates will become increasingly important indicators of research impact (*Piwowar, 2013*).

## CONCLUSIONS

We found a statistically well-supported citation benefit from open data, although a smaller one than previously reported. We conclude there is a direct effect of third-party data reuse that persists for years beyond the time when researchers have published most of the papers reusing their own data. We further conclude that, at least for gene expression microarray data, a substantial portion of archived datasets are reused, and that the intensity of dataset reuse has been steadily increasing since 2003.

It is important to remember that the primary rationale for making research data available has nothing to do with evaluation metrics or citation benefits: giving a full account of experimental process and findings is a tenet of science, and publicly-funded science is a public resource (*Smith, 2006*). We also recognize that scientists may weigh a variety of both positive and negative incentives when deciding whether and how to share their data, and the potential for increasing citations is only one of these. Nonetheless, evidence of personal benefit will help as science transitions from "data not shown" to a culture that simply expects data to be part of the published record.

## ACKNOWLEDGEMENTS

The authors thank Angus Whyte for suggestions on study design. We thank Jonathan Carlson and Estephanie Sta. Maria for their hard work on data collection and annotation. Michael Whitlock and the Biodiversity Research Centre at the University of British Columbia provided community and resources. We are grateful to everyone who helped with access to Scopus, particularly Andre Vellino, CISTI, Tom Pollard, and friends at the British Library. Finally, the authors thank their peers for feedback on preliminary and preprint versions of this manuscript. The methods and results were previously discussed on an author's blog (e.g., http://researchremix.wordpress.com/2012/07/16/many-datasets-are-reused-not-just-an-elite-few/) and in presentations (e.g., http://dx.doi.org/10.7287/peerj.preprints.14v1 and http://www.slideshare.net/tjvision/vision-ievobio12), and were published online as a preprint manuscript (http://dx.doi.org/10.7287/peerj.preprints.1v1). The first paragraph of the introduction is verbatim from *Piwowar, Day & Fridsma (2007)*; its original publication was under a CC-BY license. Publication references are available in a publicly-available Mendeley group to facilitate exploration (http://www.mendeley.com/groups/2223913/11k-citation/papers).

### Funding

This study was funded by DataONE (OCI-0830944), Dryad (DBI-0743720), and a Discovery grant to Michael Whitlock from the Natural Sciences and Engineering Research Council of Canada. The funders had no role in study design, data collection and analysis, decision to publish, or preparation of the manuscript.

### Grant Disclosures

The following grant information was disclosed by the authors:

DataONE: OCI-0830944.

Dryad: DBI-0743720.

Discovery grant to Michael Whitlock from the Natural Sciences and Engineering Research Council of Canada.

### Competing Interests

Todd Vision is an Academic Editor for PeerJ. Heather Piwowar is a cofounder of ImpactStory, a nonprofit startup that provides altmetrics data to PeerJ.

### Author Contributions

- Heather A. Piwowar conceived and designed the experiments, performed the experiments, analyzed the data, contributed reagents/materials/analysis tools, wrote the paper.
- Todd J. Vision wrote the paper, contributed to study design.

### Data Deposition

The following information was supplied regarding the deposition of related data:

In Dryad at http://doi.org/10.5061/dryad.781pv, and also https://github.com/hpiwowar/citation11k/tree/master/analysis.

### Supplemental Information

Supplemental information for this article can be found online at http://dx.doi.org/10.7717/peerj.175.

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
