# Peer review of "Data reuse and the open data citation advantage"

_PeerJ, doi:10.7717/peerj.175_

## Round 0.1 · original submission · Minor Revisions

Both reviewers think the paper is worthy of publication but needs some revisions. One reviewer suggests the second part (patterns in data reuse) of the paper could be shortened and more information on this aspect may need to be provided in the Introduction. Please try to address the reviewers' concerns about "open access citation benefit" and the usage of some terms including the maybe too positive "boost". Both reviewers raise questions about tables and figures used in the manuscript. I suggest the authors fully address the questions from the reviewers, and think about their thoughtful comments, some of which may be beyond the current manuscript.

Reviewer 1 ·

Basic reporting

The aims of the paper are to : a) assess whether making datasets public and available at time of article publication increases citations to the article and (b) describe the pattern of data reuse once data are made public. It does so using studies based on gene expression microarray data. The objectives of the paper are highly relevant since a clear demonstration that making data widely available provides authors with a citation benefit would be a strong incentive for scientists to do so.
The paper is well organized and clear, but the second objective was less convincingly introduced than the first one, although it makes an important part of the manuscript (especially in terms of number of figures). I found this second part a bit lengthy, and I suggest that the authors shorten it by removing data and/or figures (for example, Fig. 5 and 6) give the same basic information, and one of the two could be deleted while giving only results for the second one in the text.
Isn’t the term “boost” a bit too strong for a 9-10% higher citation rate? Wouldn’t “benefit” be more appropriate?
Some minor points to modify in the text and/or figures:
- p. 2, line 11: delete one of the “characterized”
- p. 2, line 42: this repeats the sentence of p. 2 lines 13-17
- p. 3, line 2: Table 1 does not show the content discussed in the text (attributes that correlate with citation rates);
- p. 3, line 14: give a reference for MeSH terms
- p. 3, line 37: add "they" between "because" and "included"
- Fig. 2 and Table 3 give the same information: remove Table 3
- p 7, line 16: chage "two" by "to"

Experimental design

The first aim is addressed using a data set of 10,555 papers screened using full-text query method, but I could not find the number of papers in each category: with or without publicly available data. This information should be provided.
- p. 4, line 7 : why switch to a different source of data here (use of ISI web of knowledge instead of Scopus: p. 2, lines 17-21)
The second aim (data reuse) is addressed using direct mentions to the data sets in papers where these are mentioned.
Overall, the methods of data collection from the various sources and the treatment of data appear to have been correctly conducted.

Validity of the findings

The analyses show that papers making microarray data sets publicly available are, on average, cited 9% more than those which do not. This impact is apparent only at least three years after publication. This finding is interesting, although the effect is weaker than previously shown with a more restricted data set. The trend found for data reuse shows that original authors tend to use their own data sets until approximately two years after these are published, while the use by third party authors tend to increase after these two years. The study also show that the number of data sets reused in any one paper tends to increase with time: most papers reused one or two data sets until 2005, and this number increases substantially thereafter.
I found the last part of the result section (p. 8, lines 1-18) and figures 8 and 9 a bit overdone, all the more that this aspect of the paper is not properly introduced ealier, especially in the introduction.

Additional comments

I would recommend shortening the second part of the result section pertaining to data reuse, which increases substantially the amount fo data presented in the paper and tends to dilute its main messages. Given that this aspect is only poorly introduced, shortening this part would certainly strengthen the paper.

·

Basic reporting

I am not qualified to comprehensively review the methods and statistical analysis but my reading raised the following:

In the ‘Primary analysis’ section, is a table mislabelled or missing? The authors state:
“We used a subset of the 124 attributes from (Piwowar, 2011d) previously shown or suspected to correlate with citation rate (Table 1).”
I expected to see the list of attributes but Table one is headed: “Proportion of sample published in most common journals”
If so, this needs to be resolved before publication. I am also interested in the evidence on which the correlates are based. The authors mention the open access citation advantage a few times in their paper, for example. Have they assumed that open access papers receive more citations in the present analysis? The open access citation benefit is much debated. Furthermore, my understanding of the cited paper by Craig et al 2007 is that it documents *decreasing* evidence of a citation benefit for open access articles. I am not aware of a systematic review of the impact of open access on citations, but a bibliography of studies, which the authors may find relevant, is here http://opcit.eprints.org/oacitation-biblio.html

Experimental design

See above.

Validity of the findings

The results of this paper are welcome evidence of the benefits to individual researchers for sharing their research data. They should and will likely be used to develop further support and policy development for data sharing and will be much discussed on social media. However, I feel the authors should tone down the positivity of the language used to describe the citation increase in their conclusions (“robust”; a “boost”, etc), given the citation benefit is just 9%.

I wondered if the authors could comment on if and how much their results could be generalised to other areas of research. Also, where should the research community’s priorities be for establishing citation or other benefits to individuals from sharing their research? Which fields? Which benefits? We know, for example, that clinical trialists are willing to share their data but practical issues and fears over inappropriate reanalysis are more important barriers to sharing –than lack of individual incentives/citations (Rathi et al. http://dx.doi.org/10.1136/bmj.e7570).

Additional comments

I assume the opening quote is not intended to add colour to paper but to provide the opening part of the introduction. As such, I found the author quoting themselves at this length from a 6 year old study was not the best way to open a paper.

Throughout the manuscript the authors refer to “open data” meaning, I assume in this context, data which are freely accessible on the web. However, there is growing recognition that “open” data means data which are available under legal terms – a license or waiver – which permit sharing and reuse with the minimum of barriers. Open data is about more than just accessibility. “Open” data must be free to “download, copy, analyse, re-process, pass them to software or use them for any other purpose without financial, legal, or technical barriers other than those inseparable from gaining access to the internet itself” [http://pantonprinciples.org/] . To my knowledge the authors are aware of the appropriate use of “open” in open data and open access and I recommend they use an alternate term to “open data” in the present manuscript to avoid engendering confusion about the term “open data”.

I’m not convinced the account of the authors’ problems in obtaining access to Scopus should be included in the current manuscript. It seems tangential to thrust of the manuscript and its analysis and would be more appropriate for a blog (I believe some of this was described on Piwowar’s blog) or commentary, and would make the manuscript more focused. Also, elsewhere in the paper the authors note that Scopus now has an API for programmatic access. The authors have previously written about the problems faced by researchers in gaining access to the literature for research – text mining, citation analysis – elsewhere.

I noticed a few typos (this is not an exhaustive list) e.g. “The lack of conventions and tool support for data Attribution”; “Which types of data reuse are facilitates by robust data standards and which types are unaffected?”

I am aware one of the authors has previously discussed publicly the present study’s data and preliminary results (http://researchremix.wordpress.com/2012/07/16/many-datasets-are-reused-not-just-an-elite-few/). Open science should not preclude papers being published, and the fact the authors discussed the data from this study before submission of their paper should have no influence on the paper’s acceptance or rejection. However, best practice regarding potentially duplicate or overlapping publications would be to state in the paper and cite the previous publications, including blogs, posters etc, in the submitted manuscript.

Recommendation to PeerJ staff regarding competing interests statements from authors. Please include these in the review version of the manuscript.

---

## Round 0.2 · accepted · Accept

I appreciate the detailed responses from the authors. The issues raised by reviewers have been appropriately addressed. I am happy to accept the manuscript.